# Structural Diversities and Phylogenetic Signals in Plastomes of the Early-Divergent Angiosperms: A Case Study in Saxifragales

**DOI:** 10.3390/plants11243544

**Published:** 2022-12-15

**Authors:** Shiyun Han, Hengwu Ding, De Bi, Sijia Zhang, Ran Yi, Jinming Gao, Jianke Yang, Yuanxin Ye, Longhua Wu, Xianzhao Kan

**Affiliations:** 1Anhui Provincial Key Laboratory of the Conservation and Exploitation of Biological Resources, College of Life Sciences, Anhui Normal University, Wuhu 241000, China; 2College of Landscape Engineering, Suzhou Polytechnic Institute of Agriculture, Suzhou 215000, China; 3CAS Key Laboratory of Soil Environment and Pollution Remediation, Institute of Soil Science, Chinese Academy of Sciences, Nanjing 210008, China; 4The Institute of Bioinformatics, College of Life Sciences, Anhui Normal University, Wuhu 241000, China

**Keywords:** Saxifragales, angiosperms, plant evolution, plastome diversity, pttRNAs, microstructural changes, gene loss, intron loss, phylogeny

## Abstract

As representative of the early-divergent groups of angiosperms, Saxifragales is extremely divergent in morphology, comprising 15 families. Within this order, our previous case studies observed significant structural diversities among the plastomes of several lineages, suggesting a possible role in elucidating their deep phylogenetic relationships. Here, we collected 208 available plastomes from 11 constituent families to explore the evolutionary patterns among Saxifragales. With thorough comparisons, the losses of two genes and three introns were found in several groups. Notably, 432 indel events have been observed from the introns of all 17 plastomic intron-containing genes, which could well play an important role in family barcoding. Moreover, numerous heterogeneities and strong intrafamilial phylogenetic implications were revealed in pttRNA (plastomic tRNA) structures, and the unique structural patterns were also determined for five families. Most importantly, based on the well-supported phylogenetic trees, evident phylogenetic signals were detected in combinations with the identified pttRNAs features and intron indels, demonstrating abundant lineage-specific characteristics for Saxifragales. Collectively, the results reported here could not only provide a deeper understanding into the evolutionary patterns of Saxifragales, but also provide a case study for exploring the plastome evolution at a high taxonomic level of angiosperms.

## 1. Introduction

Throughout the investigation of plant phylogeny, plastid data play an important role in plant systematics. As one of the most iconic features of plant cells [1], the plastome can be highly informative in reflecting the plant evolution [2]. In higher plants, the plastid is uniparentally inherited and its genome commonly possesses a quadripartite structure, 120–160 kb in size [3]. Furthermore, the substitution rates of plastomic genes lie in the middle between mitochondrial and nuclear substitution [4], although they may vary across different taxa or among genes [5,6,7]. Consequently, these features mentioned above make plastomes a perfect choice for inferring plant phylogeny [8,9]. With the advent of deep sequencing technology, plastome data has gradually served as a valuable routine tool for molecular evolutionary analyses. To date, more than 8600 plastomes are accessible via Genbank.

For angiosperms, there is a general consensus that the organization and gene content of plastomes is greatly conserved [10,11,12,13]. However, along with the rapidly expanding availability of sequence data, many variation events of plastomes have been detected in structures. For instance, total losses of 26 genes and 8 introns of plastomes were observed during the evolution of angiosperms [14]. These losses might be attributed to the transfer of plastomic genes to the nucleus or their function replacement by nuclear genes [15,16,17,18,19]. Additionally, several gene and intron losses, including *infA*, *rpl32*, *rps16*, the 2^nd^ intron of *ycf3,* etc., were proved to be highly lineage-specific among the investigated species of Malpighiales [20]. In addition, introns of plastomes, with considerable structured sequences, have been proven to undergo complicated evolution processes, resulting in either sequence conservation or mutational hotspots [21]. Most interestingly, some indels of group II introns, such as the *petD* intron, have been found to be highly useful in the phylogeny of basal angiosperms [22].

Indeed, over the processes of nucleotide mutations, notable structural changes are observed within plastomes. The most conspicuous example is the expansions or contractions of inverted repeats (IRs: IRa and IRb), which could influence not only the overall size of plastome [23,24,25,26], but also the plastomic gene organization [27,28,29]. It is also worth noting that such structural changes have taxonomic significance and can be employed as tools for evolutionary interpretation [30,31,32]. For instance, a unique pattern of *rps19* at the junction of LSC and IRb has been documented by several studies in Crassulaceae (Saxifragales) [1,33], which still holds true in a wider sampling of 69 species from this family [34]. Furthermore, a recently discovered structural variation has been attracting attention. As we know, transfer RNA (tRNA), with a characteristic clover leaf-like structure, acts an indispensable part during protein synthesis [35]. However, some variations have been identified by recent studies. More importantly, two current studies from our group strongly suggest that some novel structural variations detected in plastomic tRNAs (pttRNA) might have phylogenetic implications and can be used in potential plant barcoding [34,36].

As a representative of the early-divergent groups of angiosperms [37,38,39,40], the order Saxifragales comprises 15 families and was found to be monophyletic [41,42,43]. Due to the rapid radiation of Saxifragales [44,45], the extreme morphological diversity (trees, herbs, succulents, etc.) posed a major obstacle in earlier phylogenetic studies. As more molecular data were employed, the interfamilial relationships within Saxifragales have been generally well-resolved, with seven primary clades [34,45,46,47]. Saxifragales can serve as a good case for studying the diversification patterns in angiosperms [47]. By far, through careful investigation, several structural variations of plastomes have been observed among species of Saxifragales, such as the losses of *infA* and *rpl32* in Paeoniaceae [1,48], the intron losses of *rps16* in the genus *Penthorum* [1], the intron losses of *rpl2* in the genera *Saxifraga* and *Heuchera* [1,49], and the family-specific pattern of the IR junction and the phylogenetic informative pttRNAs’ structures in Crassulaceae [34]. In fact, structural variations of plastomes have not been comprehensively and systematically studied within this order so far. Hence, more efforts are needed to address this issue.

To explore the plastomic variations within Saxifragales, we retrieved sequence data from 208 taxa representing 11 families of this order. By comprehensive analyses, we tried to address (1) the gene content variations among the plastomes of Saxifragales, (2) the microstructural changes within introns of plastomic genes, (3) the structural diversifications of pttRNAs, and (4) the patterns of plastomic homoplasy features of this order. Thereby, our findings may facilitate a deep understanding of the evolutionary patterns of Saxifragales, and also provide an excellent case study for plastome evolution at a higher taxonomic level.

## 2. Results

### 2.1. Overall Variations of Plastomic Gene Organization among Saxifragales

To obtain a more comprehensive estimation for the plastomic features among Saxifragales, we investigated a total of 208 taxa, which are summarized in Table 1. In general, these plastomes demonstrated a rather conserved pattern, including quadripartite structure, gene order, and GC content, etc. However, several differences were still recognized with respect to genomic size, loss event for both of PCGs (protein-coding gene) and introns, and gene content at IR junctions.

Primarily, among the 208 plastomes, the ranges of 145,737 (*Crassula perforate*)–160,861 bp (*Altingia excelsa*) and 36.48 (*Myriophyllum aquaticum*)–38.55% (*Paeonia brownii*) were identified for genome size and GC content, respectively. Notably, with the whole plastome length ranging from 160,401 to 160,861 bp, Altingiaceae (represented by seven taxa) generally exhibits the largest sizes within this order (160,641 ± 216 bp), followed by Daphniphyllaceae and Iteaceae (represented by only one species). As for the GC content, the seven Haloragaceae plastomes featured the lowest value, with an average of 36.73 ± 0.22 % for total genome and 30.20–30.90% for SSC regions. Moreover, several loss events were detected for both plastomic genes and introns. The genes *infA* and *rpl32* were found lost in all the involved 20 Paeoniaceae plastomes. Intron losses were also identified in three genes, i.e., *atpF* in *Pachyphytum compactum*, *rps16* in *Penthorum chinense*, and *rpl2* in Saxifragaceae (containing 50 taxa).

In addition, we also compared the expansion and contraction of IR junctions to the adjacent genes among Saxifragales. Interestingly, as listed in Appendix A, both similarities and differences were revealed. By thorough comparison, two features were highly conserved in all 208 plastomes: (1) IRa possessed an expansion to *ycf1* (ranging from 400–1870 bp), accordingly resulting in a partial fragment (pseudo-copy) in IRb; and (2) the 3’-terminal 3 bp of *trnH-GUG* was located in IRa. On the contrary, considerable diversities were harbored by the contraction and expansion to genes of JLB (junction IRb/LSC) and JSB (junction IRb/SSC). For simplicity, six different patterns could be outlined: (1) expansion for JLB to *rps19* and contraction for JSB to *ndhF* (Altingiaceae, Daphniphyllaceae, Grossulariaceae, and Iteaceae), (2) contraction for JSB to *rps19* and expansion for JSB to *ndhF* (Penthoraceae), (3) expansion/contraction to *rps19* for JLB and only expansion for JSB to *ndhF* (Haloragaceae), (4) only expansion (Crassulaceae), or (5) only contraction (Cercidiphyllaceae), or (6) expansion/contraction (Hamamelidaceae, Paeoniaceae and Saxifragaceae) for both junctions of JLB and JSB to these two genes.

Most notably, family-specific features were found in two families: (1) the IRb regions of Crassulaceae all expanded 110 bp to *rps19*, with only four exceptions (105 bp for *Orostachys fimbriata*, *Phedimus kamtschaticus*, *P. spurius,* and *P. takesimense*); and (2) the IRa regions of Paeoniaceae all have 1077-bp expansions to *ycf1*, except for *Paeonia emodi* (1105 bp) and *P. lactiflora* (1081 bp).

### 2.2. Hypervariable Loci Assessment among Saxifragales

To reach an overall exploration of the sequence variations among Saxifragales plastomes, the DNA polymorphism was evaluated by nucleotide diversity values (π). The comparative analyses were performed in ten groups: nine separate families (excluding Iteaceae and Penthoraceae, that with only one sequence data released, respectively) (Figure 1c), and the whole order of Saxifragales. As detailed in Appendix A, the patterns of hypervariable loci generally exhibited a high degree of diversity among the families. As for the number of HPR (highly polymorphic region), Crassulaceae and Saxifragaceae displayed the smallest (three for both), followed by Haloragaceae and Hamamelidaceae with five loci, respectively, while the highest was seen for Altingiaceae (14), followed by Daphniphyllaceae and Cercidiphyllaceae. Excluding the same loci among different groups, we identified a total of 55 HPRs. Of these HPRs, nine (16.4%) and six (10.9%) were located in SSC and IR regions, respectively, with the majority (72.7%) in LSC regions (Figure 1a). Moreover, ten separate HPRs were shared by some families. For instance, one HPR (*trnR-ACG-trnN-GUU-ycf1*) was possessed by three families (Altingiaceae, Paeoniaceae, and Grossulariaceae). Each of the remaining nine HPRs were observed in two of the families.

Most interestingly, the patterns of π values appeared to be the opposite. As outlined in Figure 1b, the four families (Saxifragaceae, Crassulaceae, Haloragaceae, and Hamamelidaceae) with less HPR harbored much higher values—averaging from 0.03294 to 0.08285—than the other five (Grossulariaceae, Paeoniaceae, Daphniphyllaceae, Altingiaceae, and Cercidiphyllaceae) with more loci present as the lower values (0.00417–0.02153). Further, at the order level, only two regions were classified as HPRs, i.e., *rpoB* (π = 0.13251) and *ndhH-rps15-ycf1* (π = 0.16323).

### 2.3. Microstructural Changes within Plastomic Introns

In total, 432 indel events have been obtained from the aligned intron matrixes of 17 separate plastomic genes (five tRNA genes and 12 PCGs) in the 208 Saxifragales taxa (Appendix A). Among them, 221 indels (51.2%) represented deletions, while the remaining 211 (48.8%) were insertions. Moreover, notable heterogeneities of indel sizes have been observed (ranging from 1 to 176 nt), with the largest proportion of 5-nt indels (14.7%), followed by 6-nt (11.6%) and single-base (9.3%). However, overall, these indels were mostly no more than 10 nt in size (79.7%). Additionally, the indels remarkably displayed uneven distributions among the 17 intron-containing genes. In general, the greatest number of indels were harbored by *trnK-UUU* (62, 14.4%), then by *ndhA* (40, 9.3%), and *rps16* (33, 7.6%). In contrast, three plastomic genes, i.e., *rps12*, *ndhB,* and *rpl2*, showed far fewer indels, with only one (0.2%), four (0.9%), and five (1.2%), respectively.

Strikingly, the indel patterns among Saxifragales exhibited a high potential for assessing phylogenetic relationships at the family level. With the only exception of Hamamelidaceae, all investigated families had multiple family-unique indels. As can be seen from Table 2, Paeoniaceae had the highest number of unique indels (82) compared to other nine families (3–41). Meanwhile, indels from the introns of *trnK-UUU*, *ndhA,* and *rps16* could successfully distinguish seven families, respectively. Notably, the 3′ introns of *rps12* are highly conserved across Saxifragales, with no family-unique indels. Further, in the intron regions of *ndhB* and *trnL-UAA*, only one unique indel site was observed in Iteaceae and Crassulaceae, respectively.

### 2.4. Specific Markers from pttRNAs’ Structural Diversifications

The secondary structures of pttRNAs were predicted within Saxifragales. Among the 7488 pttRNAs investigated in this study, 10 putative non-typical structures were observed for all the constituent families (Figure 2a). In general, four separate categories can be inferred from these unconventional pttRNA structures, including (1) an additional loop at AC-arm (tRNA^Arg^-ACG with a 4-nt loop, tRNA^Thr^-UGU with a 2-nt loop); (2) an expanded 9-nt ANC-loop (tRNA^Val^-UAC and tRNA^Leu^-UAA); (3) a long variable region at V-arm (tRNA^Leu^-CAA, tRNA^Ser^-UGA, tRNA^Ser^-GCU, tRNA^Ser^-GGA, and tRNA^Tyr^-GUA); and (4) an extra loop at Ψ-arm (tRNA^Cys^-GCA), respectively.

To gain deeper insights into the implications harbored by such heterogeneities, overall comparison was further conducted among all the non-typical pttRNA structures mentioned above. Strikingly, we found that these structures could be highly specific to different hierarchies of Saxifragales (i.e., interfamily and intrafamily levels). As for intrafamilial level, nine families (except for Iteaceae and Penthoraceae), which contained more than one sequenced plastome, were considered for further analyses. Among the nine families, three were examined to be strictly conserved for the non-typical structures, while substantial variations were identified in the other six families. Of these, four of the six had significant infra-family differences (tRNA^Thr^-UGU for Haloragaceae, tRNA^Ser^-UGA for Hamamelidaceae, and tRNA^Val^-UAC for Daphniphyllaceae and Grossulariaceae) (Figure 2b). For instance, within the family Grossulariaceae (representing five *Ribes* taxa), two types of ANC-loop in plastomic tRNAVal-UAC were revealed: (1) type A (typical 7-nt loop) for *R. nevadense* and *R. roezlii* and (2) type B (expanded 9-nt loop) for the other three taxa. Most surprisingly, in contrast to other families, we found that Crassulaceae and Saxifragaceae harbored wider intra-family variations, with six and four types of pttRNAs, respectively.

Interestingly, at the interfamilial level, unique structural patterns were also detected for five families through three pttRNAs. Figure 2c summarized the details of the characteristic features: (1) differing from other 10 families, Daphniphyllaceae featured 5′-GAAUAA-3′ in the V-loop of tRNA^Ser^-UGA, (2) for tRNA^Ser^-GCU, the distinctive 5′-UUA-3′ loop of variable regions was merely identified in all 20 Paeoniaceae taxa, (3) Crassulaceae was the only one that evolved five different types of V-loop in tRNA^Ser^-GCU, which were not found in the other families, (4) 5′-UU-3′ at AC-loop of tRNA^Thr^-UGU was unique for Altingiaceae, and (5) 5′-GG-3′ at Ψ-loop (*Glischrocaryon aureum*) or a non-novel structure (all the remaining six accessible plastomes) of tRNA^Thr^-UGU was observed in Haloragaceae.

### 2.5. Phyloplastomic Analyses among Saxifragales

To explore the taxonomic relationships within Saxifragales, 79 plastomic PCGs and complete cp genomes (CPGs) from 208 ingroups and three outgroups (Rosids) were employed for analyses, respectively. The phylogenomic inferences, obtained by 77,274-bp (PCG dataset) and 262,199-bp (CPG dataset) concatenated matrixes, respectively, generally yielded highly similar topologies between ML and BI trees (Figure 3 and Appendix A).

Significantly, our results recovered two major clades of Saxifragales: (1) Paeoniaceae plus woody group (BS = 97%, PP = 0.99), and (2) “core Saxifragales” (BS = 100%, PP = 0.99). Therein, woody group comprises four families: Altingiaceae + (Daphniphyllaceae + (Cercidiphyllaceae + Hamamelidaceae)) and is sister to Paeoniaceae with relatively high support. In addition, core Saxifragales could be additionally divided into two alliances: Crassulaceae alliance (BS = 100%, PP = 0.99), with Crassulaceae sister to Haloragaceae + Penthoraceae, and Saxifragaceae alliance (BS = 100%, PP = 1.0), including Iteaceae + (Saxifragaceae + Grossulariaceae). Overall, our results were highly congruent with those of former analyses [34,46,47,48]. In general, the 79-PCGs datasets allowed high resolution and support for all the interfamilial relationships, except for those of two clades, i.e., Cercidiphyllaceae + Hamamelidaceae (BS = 65%, PP = 0.91) and Daphniphyllaceae + (Cercidiphyllaceae + Hamamelidaceae) (BS = 55%, PP = 0.92).

Significantly, by further exploration, with the exclusion of Iteaceae and Penthoraceae (represented by one species, respectively), the monophyly of each of the nine families investigated was strongly supported (BS = 100%, PP = 1.0). Moreover, within these families, most of the primary subclades were also well resolved to be monophyletic (BS = 100%, PP = 0.99 or 1.0), such as Sempervivoideae and Kalanchoideae in Crassulaceae; Hamamelidoideae in Hamamelidaceae; as well as *Peltoboykinia* (including *Chrysosplenium* and *Peltoboykinia*), *Darmera* (including five genera such as *Mukdenia* and *Oresirophe*), and *Heuchera* (including *Heuchera* and other related four genera) groups in Saxifragaceae, etc. Moreover, several genera turned out to be non-monophyletic, e.g., *Altingia* and *Liquidambar* (Altingiaceae), *Sedum*, *Hylotelephium* and *Orostachys* (Crassulaceae), and *Heuchera*, *Tiarella,* and *Mitella* (Saxifragaceae).

## 3. Discussion

In the present study, a thorough and comprehensive comparison among 208 Saxifragales plastomes was provided with two main aims: (1) reaching an overall understanding of the plastomic evolutionary diversities within this order; and (2) assessing the potential barcoding performance of plastid genomes for not only phylogeny, but also the specific evolved characteristics in plant. Notably, both sequence and structure diversities were clarified across these plastomes, including basic genomic features, gene content, IR boundary patterns, DNA polymorphisms, introns variabilities, structural variations in pttRNAs, and phylogenetic signals and interpretations. Collectively, the results presented here would provide further insights into the evolutionary elucidations for the early-divergent angiosperms, especially for the order Saxifragales.

Despite the quite conserved gene content of angiosperm plastomes, loss events were not rare across their evolutionary history, supposedly after the first endosymbiotic event [14,50]. Jansen et al. [14] sampled 77 PCGs and four rRNAs from each of the 64 plastomes, representing the most major angiosperm groups, and remarkably, this study suggested that all loss events were identified in monocot and eudicot lineages. As one of the largest orders in the core eudicots clade [43], Saxifragales was also found to embody several losses for two genes and three introns of plastomes in our study. Therein, all the 20 investigated Paeoniaceae plastomes lost *infA* and *rpl32*, reinforcing the results of our previous works [34,48]. In fact, over the plastomic evolution, independent losses of *infA* have occurred repeatedly among angiosperms [14,51,52]. Additionally, in these cases, a documented mechanism, lateral gene transfer from plastome to nuclear, has been described for the losses of *infA* in Rosids [53] and *rpl32* in Salicaceae [54,55]. Likewise, the observed three intron losses (*atpF*, *rps16,* and *rpl2*) were also present multiple times [14,49], such as the intron losses of *atpF* in Malphigiales [56], *rps16* in Celastraceae [57], and *rpl2* in Lythraceae [58], etc. It is worth noting, by far, that there are three possible pathways for these involved plastomic intron losses: (1) recombination of the RNA-edited intron lacking copy and the initial intact copy (*atpF*) [56,59]; (2) homologous recombination and reverse transcript mediated mechanism (*rps16*) [57]; and (3) unequal crossover and gene conversion (*rpl2*) [60,61,62].

As in earlier studies of molecular phylogeny, only several plastid gene loci were employed, such as *matK*, *rbcL*, *trnH-psbA*, etc. However, some of these core markers have been documented to display low efficiency in resolving many closely related taxa [23,63,64,65,66]. Over the years, as the organelle genomics progressed, it has been well known that plastomes embody numerous potential mutations clustering as “hotspots” across evolution [23,67,68]. Undeniably, delving into the specific HPRs in investigated taxonomic groups is necessary for reaching better barcoding performance [67]. In the case of this work, our results demonstrated the variability patterns of plastomes among the nine families of Saxifragales. Within the plastomes, we observed that LSC regions occupy the major hotspots loci, which was congruent with previous studies, including Liu et al. [69] in *Oresitrophe* and *Mukdenia* (Saxifragaceae), Liu et al. [67] in *Ormosia* (Fabaceae), and Xu et al. [70] in *Saccharum* (Poaceae). In contrast to LSC and SSC, IR regions generally could accommodate fewer mutations, which might be confined to consistency correction of its two copies [71,72].

In addition, we further assess the applicability of the identified HPRs. Significantly, with the exception of a few uniform sites, apparent substantial inconsistencies were found in comparison with former studies. For example, the results from this work (with 33 taxa) and Wang et al. [73] (with 6 taxa) identified five and seven HPRs in Hamamelidaceae, respectively, only sharing one identical loci (*ndhG*). Moreover, a similar finding was also found in Grossulariaceae taxa [74]. According to Shahzadi et al. [23], these discrepancies might be caused by different diversity levels of the involved taxa or the impact of analysis approaches. Thus, wider samples and more efforts are needed to clarify the hotspots’ patterns within Saxifragales. Most importantly, plastomic HPRs might serve as potential markers for plant DNA barcoding and phylogenetic inferences.

Intron and IGS regions of plastome sequences, to our best knowledge, could serve as primary sources of phylogeny data [75,76]. Overall, compared to most coding sequences, these two noncoding regions generally possessed higher diversity [22]. It is also worth mentioning that indels, known as the microstructural variabilities, were frequently present at these regions [75]. Yet, among Saxifragales plastomes, the distribution pattern of indels in the noncoding regions were still unclear. Here, with relatively large samples, our thorough analyses allowed determinations of 432 indels within introns. Notably, their size patterns, most no longer than 10 nt, turned out to be consistent with the results of Lohne et al. [22] in the *petD* intron and *petB-petD* region among angiosperms. Moreover, as mentioned above, our results found that the 3′-*rps12* introns had the rarest indels across Saxifragales. Interestingly, a similar finding was also found by Graham et al. [75] in 3′-*rps12* introns of 31 angiosperms. Most significantly, such highly conservative evolution of this intron might be implied by the unique characteristic of *rps12*. As the sole trans-splicing plastomic gene in plants [77], it has been documented that the unique exon division of *rps12* requires conserved intron regions [78]. Above all, by the comparative analyses of plastomes among the constituent families within Saxifragales, abundant indel events were revealed to be highly family-specific, which could well play an important role in plant family level discrimination.

Chloroplast, the metabolic center of higher plants, can involve many fundamental synthesis pathways, such as proteins, lipids, phytohormones, etc. [2]. Encoded by plastomes, pttRNAs act as indispensable parts in translations as the linkage between mRNA and proteins [36,79,80]. As presented by Brennan and Sundaralingam [81], tRNAs could be divided into two categories based on the length of the V-arm: the most common, type I, features a short V-loop (4–5 nt) and type II, with a longer region (10 nt or over), is now thought to be limited to leucine, serine, and tyrosine [36,81,82]. Interestingly, it has been proposed that the bulky V-arm might have an impact on tRNA’s function, by assisting the combination with ribosomes [82]. Several studies have recently explored pttRNAs’ structural variations from type II, and found that they might be highly conserved at relatively low taxonomic ranks, such as two genera of Crassulaceae (*Aeonium* and *Monanthes*) [34], *Viburnum* of Adoxaceae [83] and *Bletilla* of Orchidaceae [36].

Here, we provided a comprehensive investigation of pttRNAs focused on the order level. The unique characteristics of type II tRNAs were detected in five pttRNAs from three isotypes in Saxifragales (Leu, Ser, and Tyr). At first, in the current study, we identified strong phylogenetic signals in several families. For instance, by extending our previous work [34] using a larger dataset, we reconfirmed the unique 5′-AUA-3′ V-loop of tRNA^Tyr^-GUA in Kalanchoideae (Crassulaceae); all involved *Ribes* (Grossuriaceae) taxa possessed an expanded ANC-loop in tRNA^Val^-UAC, except for two closely related species (*R. nevadense* and *R. roezlii*, [BS] = 100, [PP] = 1.0) with an ordinary 7-nt loop. Then, within the order Saxifragales, our results clearly demonstrate that Crassulaceae harbors the most extensive diversifications in pttRNAs’ structures. Interestingly, compared to other families, the Crassulaceae clade had considerably longer branch lengths from the early well-supported phylogram of Saxifragales [46,48], indicating that this clade had accumulated more mutations in the process of evolution [84,85,86,87]. Finally, we compared the ANC-loop structures of tRNA^Val^-UAC and tRNA^Leu^-UAA among Saxifragales, respectively. It was notable that most of the investigated families had an expanded nine-bases loop in the two pttRNAs. However, a typical 7-nt ANC-loop was also identified, which mainly comes from pttRNAs of Saxifragaceae (with the exception of two *Saxifraga* taxa). It has been proposed that these expanded ANC-loops were attributed to the distal mismatch at the ANC-helix [88,89], perhaps C-U and A-C mismatches for Valine and Leucine, respectively. Most importantly, this unusual structure might influence the step size in translocation and result in a different reading number of bases (three or four) [89,90], which could further generate an impact on the translation. The findings reported here clearly reinforce the high potential role of pttRNA structures in plant taxonomy and DNA barcoding.

Previous efforts have been committed to exploring the backbone phylogeny of Saxifragales. Notably, there are several classic studies in resolving taxonomic relationships within this order, including Fishbein et al. [45], with three plastid and two nuclear genes; Jian et al. [46], based on ten plastomic, four mitochondrial, and two nuclear genes; and Soltis et al. [47], by supermatrix data. In general, these works have collectively resolved parts of the major nodes, such as “core Saxifragales”, Saxifragales, and Crassulaceae alliances. However, by far, the unresolved issues seem to converge upon several families, including Paeoniaceae and the members of woody group, mainly due to the poor support and varied taxonomic positions. For instance, Paeoniaceae was found variedly and weakly a sister to Crassulaceae alliance, Peridiscaceae, or woody group, etc. by different analyses.

In this study, the phyloplastomic analyses yield generally well-supported topologies among 208 Saxifragales taxa. For both the CPG and PCG trees, all the relatively well-supported (BS/PP > 90%) nodes possessed the same topologies, which occupied the majority of all the nodes. Significantly, Paeoniaceae and the woody group formed a relatively well-supported clade in the CPG tree (97% in ML and 0.99 in BI), which received a stronger support not only than the PCG tree, but also than that of our previous work (employing 83 plastomic genes) with 89% in ML and 1.0 in BI [48]. Furthermore, within the woody group, Altingiaceae was sister to (Daphniphyllaceae + (Cercidiphyllaceae + Hamamelidaceae)), differing from (Cercidiphyllaceae + Daphniphyllaceae) + (Hamamelidaceae + Altingiaceae) in Fishbein et al. [91] or (Hamamelidaceae + (Cercidiphyllaceae + Daphniphyllaceae)) + Altingiaceae in Jian et al. [46] and Han et al. [34]. However, all of these relationships received poor support. As Fishbein et al. [45] proposed, the unresolved phylogeny in Saxifragales could be partially explained by the unequal pattern of branch lengths, especially for the placement of Paeoniaceae, which possesses the longest branch within Saxifragales. Moreover, the unclear relationships among the woody group might due to an ancient, rapid radiation [46,48]. Most importantly, our phylogenies might offer an improvement in clarifying the taxonomic position of Paeoniaceae. To better resolve these long-standing enigmas in Saxifragales, further efforts are necessary.

Most interestingly, many evolutionary signals were revealed by the phylogenies in combination with the identified specific characteristics (microstructural changes and pttRNA structures). On one hand, overall, a total of 373 informative indels were marked to the relevant nodes of the PCG tree (Figure 3 and Appendix A). First, for the deep-level nodes of Saxifragales, indels were relatively rare, with only nine events: indels No. 80, 190, 199, and 207 for Crassulaceae alliance; No. 87 and 148 for Saxifragaceae alliance; No. 268 and 272 for the woody group; and No. 247 for Paeoniaceae + woody group. Meanwhile, in turn, numerous characteristics were marked at lower-level nodes, which contained not only the family-specific indels (196 indels in total, as summarized in Table 2), but also those specific to the internal nodes within families (148 indels). Examples included the subfamily Hamamelidoideae in Hamamelidaceae (indels No. 45 and 252); the group *Heuchera* in Saxifragaceae (indels No. 77, 233, 246, 279, and 324); and the genus *Rhodiola* in Crassulaceae (indels No. 140 and 280), etc. Further, the remaining 29 indels were found to be independent to several leaf nodes. For instance, indel No. 14, an 8-nt SSR insertion, was a homoplastic feature to *Disanthus cercidifolius* and *Rhodoleia championii*. In particular, these informative indels seemed to be independently accumulated among different taxa, which supported the expectation that microstructural changes were caused by various mutational processes [22].

On the other hand, the combination between phylogenies and pttRNAs’ structural diversities allowed rather clear insights into the evolutionary patterns within Saxifragales. Based on the simplified PCG tree, a total of 30 lineage-specific characteristics of Saxifragales were concluded by eight pttRNAs (Figure 4 and Table 3). Among these pttRNAs, multiple heterogeneities were observed in tRNA^Leu^-CAA (four at the V-loops), tRNA^Thr^-UGU (five at the AC-arms), tRNA^Ser^-UGA (six at the V-loops), and tRNA^Ser^-GCU (seven at the V-loops) across the order. In contrast, tRNA^Tyr^-GUA, tRNA^Ser^-GGA, tRNA^Leu^-UAA, and tRNA^Val^-UAC were found to be more conserved, with only two different structures, respectively. Most interestingly, among all the constituent families with multiple samplings, Grossulariaceae, Cercidiphyllaceae, and Daphniphyllaceae maintained the most family-specific characteristics, six for each. There might be two possible processes for such a phenomenon: one is preserving in slow-evolving species and the other is back mutations [92]. To further understand these evolutionary characteristics, more plastomes’ data and ulterior exploration are needed. Overall, the phylogenetic signals inferred here would not only offer potentially specific markers for Saxifragales taxa, but also provide further insights into their plastomic evolution.

## 4. Materials and Methods

### 4.1. Data Retrieval of Plastomes within Saxifragales

To reach the comprehensive comparison among Saxifragales plastomes, our dataset comprised all the available sequences from 208 species in 11 families, covering all seven major lineages of this order. Notably, 14 sequences were generated by our previous work (Appendix A). After retrieval from NCBI, the annotations of all the sequence data were carefully checked. Therein, GeSeq [93] and CPGAVAS2 [94] were employed to examine potential annotation errors. Further, we manually modified the resulting annotated genes by BLAST [95]. Finally, the overall map of plastomes was presented using Chloroplot [96].

### 4.2. Comparative Analyses of the Sequence Variations among the Plastomes

Subsequently, comprehensive analyses of plastomic variation were performed among the processed datasets. First, all the intron-containing plastomic genes were extracted and Bioedit was employed to select and trim the coding regions for further analyses. Then, MAFFT version 7.505 [97] was used to align the intron sequences among Saxifragales. Based on this, we made more elaborate modifications by Bioedit [98] according to the principles described by Borsch et al. [99]. For instance, the single-positional nucleotide adjacent to an entire indel was insufficient to be identified as an independent event. To avoid such misjudgments, the aligned gaps were all manually modified at the same column [22,100].

Additionally, the highly polymorphic regions (HPR) among Saxifragales plastomes were also explored at the intra- and interfamily levels. The aligned plastome sequences were, respectively, imported into the sliding window analysis of DnaSP 6 [101] using the parament settings described by Han et al. [34]. The nucleotide divergence (Pi) values were then estimated. The hotspot loci were further determined according to the criteria of Bi et al. [102].

### 4.3. Comparative Analyses of the Structural Diversifications among the Plastomes

To explore evolutionary implications of plastomic structures, the patterns of IR boundaries were analyzed by IRscope [103]. After verifying the gene annotations, we counted and compared the sizes of extension and contraction of each IR region. Moreover, all pttRNAs of samplings were extracted. Then, secondary structure predictions were performed with tRNAscan-SE v.2.0.3 [104].

### 4.4. Phyloplastomic Reconstruction among Saxifragales

For deeper insights into the taxonomic relationships within Saxifragales, phyloplastomic analyses were implemented for all taxa investigated here. Three species of Rosids, a large clade closely related to Saxifragales [41,105], were selected as outgroups. Two separate datasets were employed for analyses: (1) 79 plastomic PCGs and (2) complete chloroplast genomes (CPGs). The preparations of phylogenetic datasets were conducted by DAMBE for the retrieval of all protein-coding genes (PCGs) [106], MAFFT for alignment [97], and SequenceMatrix for concatenation [107].

After that, phylogenetic trees were built by two methods: maximum-likelihood (ML) and Bayesian inference (BI). Firstly, the ML trees were inferred with RAxML 8.2.12 [108] by conducting 50 runs and 1000 bootstrap replicates under the GTRCAT model. Moreover, we checked the bootstrap convergence by the “-I autoMRE” command in RAxML. Secondly, a Bayesian phylogenetic analysis was carried out by MrBayes 3.2.7a [109]. Under the optimal models calculated with ModelTest-NG [110], the phylogenetic inferences were generated using two independent runs, each with four Markov chains for 20 million generations (sampling every 1000 generations). Then, the convergence was confirmed by Tracer 1.7.1 [111].

## 5. Conclusions

In the current study, by relatively wide sampling, the comprehensive diversities among the 208 plastomes from 11 families of Saxifragales were thoroughly explored. Several loss events were observed from two genes and three introns, in particular, the intron loss of *atpF* in *Pachyphytum compactum* was first reported here. Then, we further investigated the gene content at IR boundaries, DNA polymorphism, indels in the introns of all 17 intron-containing genes, and the pttRNA secondary-structure diversities. Significantly, abundant phylogenetic implications were revealed from them, suggesting that they have strong potential roles in serving as specific markers for Saxifragales. Moreover, our phylogenetic interpretations, based on two datasets (CPGs and PCGs), generally well recovered the internal branching patterns in this order with high resolution. More importantly, the combined phylogenies with indels and pttRNA structural features could provide further insights into the evolutionary patterns among Saxifragales plastomes. Therein, Grossulariaceae, Cercidiphyllaceae, and Daphniphyllaceae were found to retain the most plesiomorphic features. Collectively, our results presented here will facilitate the understanding of the plastome evolution in Saxifragales, and accordingly, provide a case study for comparative plastomics among the early-divergent angiosperms.

## Figures and Tables

**Figure 1 plants-11-03544-f001:**
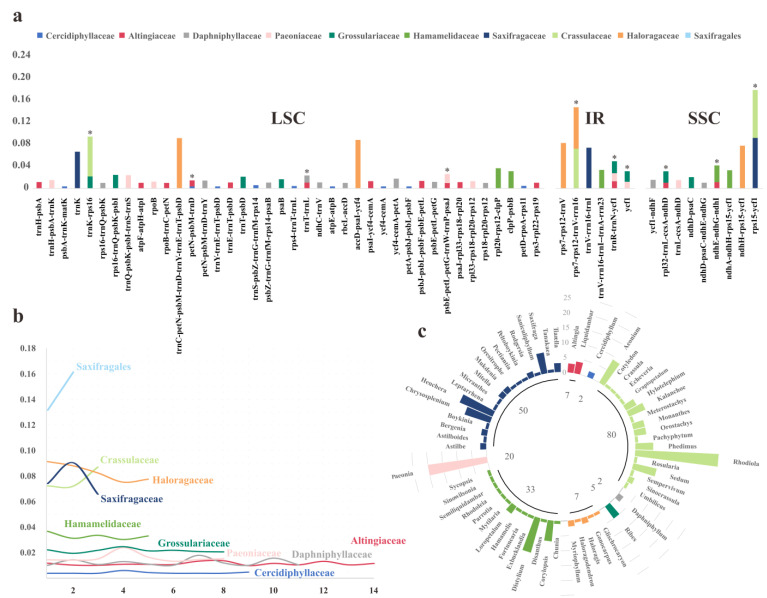
The hotspots patterns among Saxifragales. (**a**) Distributions of all the identified HPR loci, the HPRs that were shared by multiple families were marked by *; (**b**) Patterns of π values of different lineages; (**c**) Taxon samplings of this analysis, with the numbers in the circle indicating the numbers of sampling plastomes in each family, the numbers and heights of columns represented the number of genera involved, and plastomes that contained in each genus.

**Figure 2 plants-11-03544-f002:**
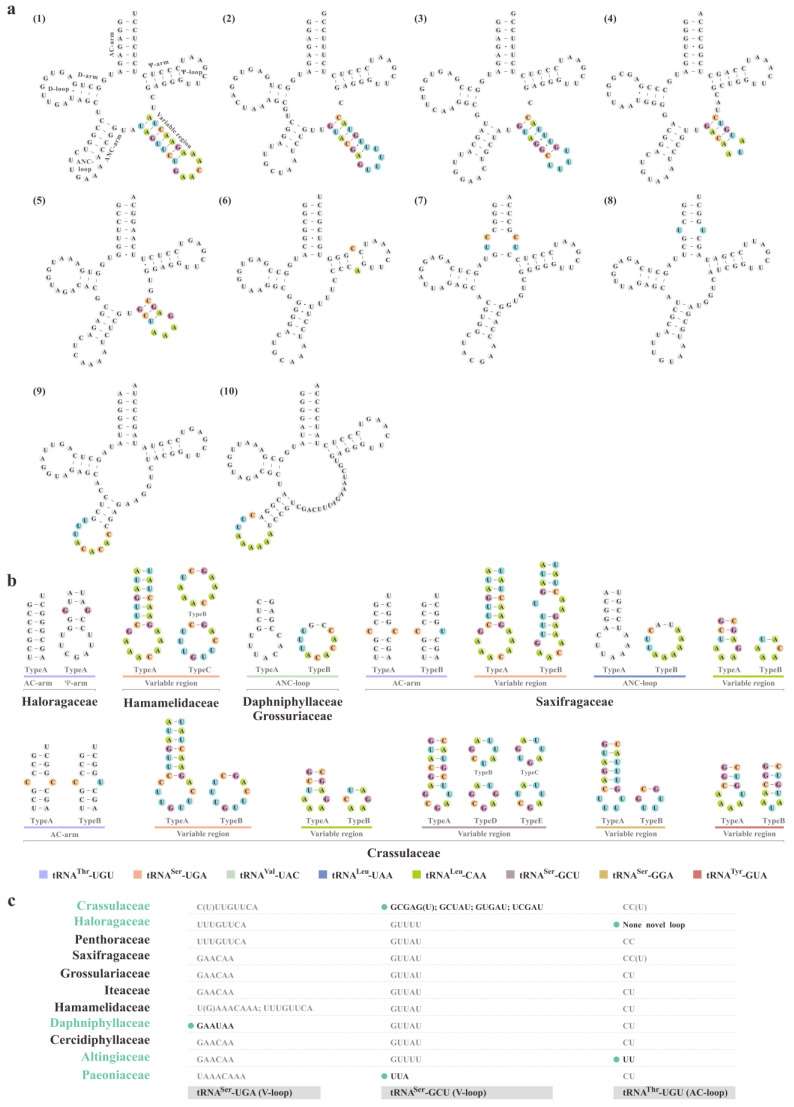
The pttRNAs’ structural diversifications among Saxifragales. (**a**) The predicted secondary structures of pttRNAs from Altingiaceae (as a representative) and the locations of non-typical structures were indicated by color:, (1) tRNA^Ser^-UGA, (2) tRNA^Ser^-GCU, (3) tRNA^Ser^-GGA, (4) tRNA^Tyr^-GUA, (5) tRNA^Leu^-CAA, (6) tRNA^Cys^-GCA, (7) tRNA^Arg^-ACG, (8) tRNA^Thr^-UGU, (9) tRNA^Val^-UAC, (10) tRNA^Leu^-UAA; (**b**) Abundant diversities identified within the 11 investigated families; (**c**) Unique structural patterns that were detected for five families through three pttRNAs.

**Figure 3 plants-11-03544-f003:**
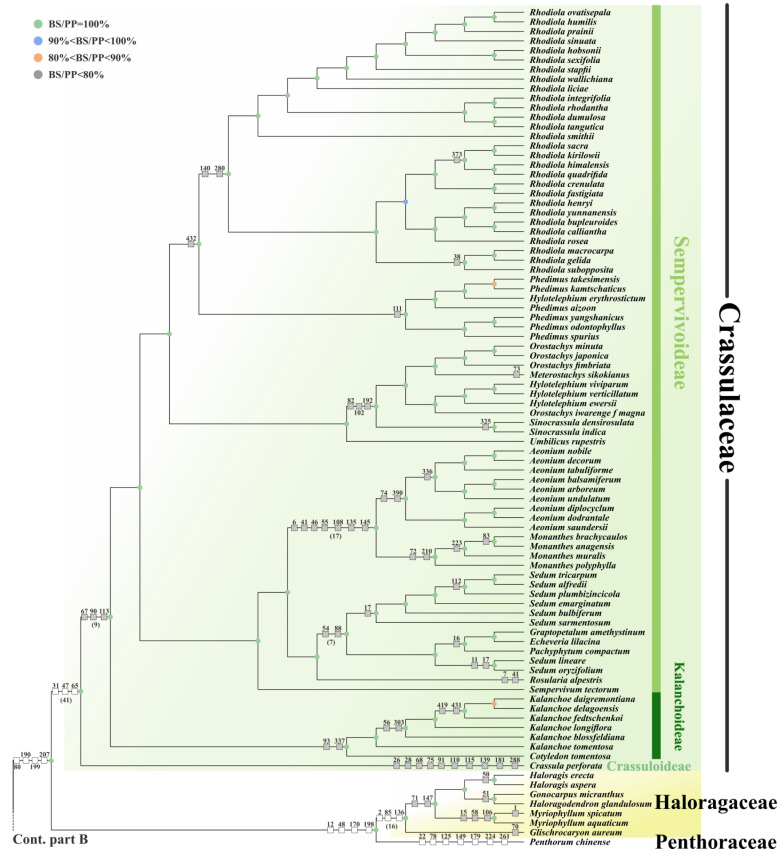
The phylogenetic tree obtained by 79 PCGs among 208 Saxifragales plastomes, with BS and PP values shown by color circles. White boxes indicate the specific indels for deep-level nodes. Lined boxes indicate the specific indels for lower-level nodes. As those clades with too many specific indels were limited in branch lengths, we marked as many indels as possible, and the total number was noted under the corresponding clade.

**Figure 4 plants-11-03544-f004:**
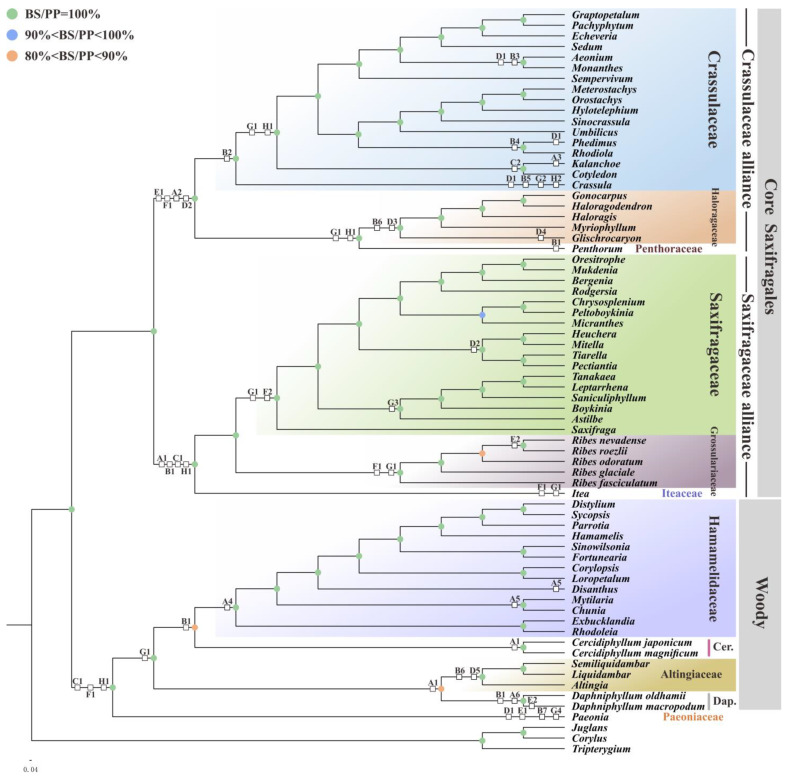
The simplified PCG tree combined with pttRNA structural characteristics. White boxes indicate the specific structures, which are detailed in Table 3. Note that the families Cercidiphyllaceae and Daphniphyllaceae are abbreviated as Cer. and Dap., respectively.

**Table 1 plants-11-03544-t001:** Basic genomic characteristics among the investigated 208 Saxifragales plastomes.

Taxa	Size (Base Pair, bp)				GC Content (%)			
	Total	LSC	IR	SSC	Total	LSC	IR	SSC
Altingiaceae	160,641 ± 216	88,882–89,162	26,274–26,471	18,917–19,011	37.93 ± 0.03	36.04–36.10	43.04–43.08	32.18–32.42
Cercidiphyllaceae	159,877 ± 32	88,035–88,058	26,427–26,434	18,973–18,965	37.92 ± 0.01	36.00	43.00	32.40
Crassulaceae	150,690 ± 1013	79,465–83,253	24,810–25,984	16,520–17,111	37.75 ± 0.10	35.45–36.29	42.80–43.31	31.09–32.40
Daphniphyllaceae	160,273 ± 192	88,075–88,103	26,546–26,605	18,970–19,095	37.86 ± 0.04	36.00–36.10	42.90–43.00	32.10–32.10
Grossulariaceae	157,559 ± 313	86,812–87,412	25,887–26,018	18,334–18,562	38.13 ± 0.02	36.20	43.08–43.14	33.20–33.40
Haloragaceae	159,050 ± 781	88,165–89,941	25,637–25,978	18,469–19,000	36.73 ± 0.22	34.20–35.00	42.73–42.88	30.20–30.90
Hamamelidaceae	159,293 ± 477	87,102–89,016	26,209–26,422	18,127–19,173	38.00 ± 0.07	35.75–36.35	43.04–43.22	32.27–32.89
Iteaceae	160,258	88,714	26,648	18,248	37.10	34.80	42.70	31.60
Paeoniaceae	152,834 ± 429	84,242–86,057	25,246–25,751	16,681–17,423	38.41 ± 0.05	36.61–36.83	43.04–43.18	32.57–33.02
Penthoraceae	156,686	86,735	25,776	18,399	37.30	35.20	42.80	31.30
Saxifragaceae	154,057 ± 2863	79,310–88,109	25,097–26,224	15,082–18,447	37.77 ± 0.19	35.05–36.22	42.69–43.28	31.16–32.85

**Table 2 plants-11-03544-t002:** Family-specific indels identified from the intron matrix of 17 plastomic genes among Saxifragales.

Taxa	Total	*trnA*	*trnI*	*trnK*	*trnL*	*trnV*	*atpF*	*clpPa*	*clpPb*	*ndhA*	*ndhB*	*petB*	*petD*	*rpl16*	*rpl2*	*rpoC1*	*rps12*	*rps16*	*ycf3a*	*ycf3b*
Alting.	10/4/14	-	1/-/1	2/1/3	-	-/1/1	2/-/2	-	-	-	-	-	1/-/1	1/-/1	-	1/-/1	-	1/-/1	-/1/1	1/1/2
Cercidi.	2/3/5	-	-	-	-	-	-	-/1/1	-	1/-/1	-	-	-/1/1	-	-	1/-/1	-	-/1/1	-	-
Crass.	26/15/41	-	1/-/1	2/1/3	1/1/2	-/1/1	-	2/2/4	5/-/5	3/1/4	-	1/1/2	0/2/2	1/1/2	1/-/1	-/1/1	-	5/1/6	2/2/4	2/1/3
Daphni.	1/2/3	-	-	-/1/1	-	-	-	-	-/1/1	-	-	1/-/1	-	-	-	-	-	-	-	-
Grossu.	6/6/12	-	1/-/1	-	-	1/-/1	-/1/1	-	-	-/1/1	-	-	3/-/3	1/2/3	-	-	-	-/1/1	-	-/1/1
Halora.	5/11/16	-/1/1	-	1/-/1	-	-	-/2/2	1/-/1	-	2/3/5	-	-	-	-	-	-/2/2	-	-/1/1	1/1/2	-/1/1
Hama.	-	-	-	-	-	-	-	-	-	-	-	-	-	-	-	-	-	-	-	-
Itea.	2/8/10	-	-	-/2/2	-	-	-	-	-	-/1/1	-/1/1	-	-	1/1/2	-	-/1/1	-	-/2/2	-	1/-/1
Paeonia.	43/39/82	1/-/1	3/2/5	3/4/7	-	2/2/4	2/6/8	4/3/7	2/3/5	5/1/6	-	-	5/4/9	3/4/7	-/1/1	2/3/5	-	6/4/10	3/1/4	2/1/3
Pentho.	2/5/7	-	1/-/1	-/1/1	-	1/-/1	-/1/1	-/1/1	-	-/1/1	-	-/1/1	-	-	-	-	-	-	-	-
Saxifra.	4/2/6	-	-	-	-	-	-	-	3/-/3	-	-	-/2/2	1/-/1	-	-	-	-	-	-	-

**Table 3 plants-11-03544-t003:** The evolutionary signals identified from the structures of pttRNAs among Saxifragales.

Types of pttRNAs		Specific Structures	
A	tRNA^Ser^-UGA (V-loop)	1	5’-GAACAA-3’
		2	5’-UUUGUUCA-3’
		3	5’-CUUGUUCA-3’
		4	5’-GAAACAAA-3’
		5	5’-UAAACAAA-3’
		6	5’-GAAUAA-3’
B	tRNA^Ser^-GCU (V-loop)	1	5’-GUUAU-3’
		2	5’-GCGAU-3’
		3	5’-UCGAU-3’
		4	5’-GCUAU-3’
		5	5’-GUGAU-3’
		6	5’-GUUUU-3’
		7	5’-UUA-3’
C	tRNA^Tyr^-GUA (V-loop)	1	5’-AUA-3’
		2	5’-AAAAU-3’
D	tRNA^Thr^-UGU (AC-loop)	1	5’-CU-3’
		2	5’-CC-3’
		3	No additional loop
		4	5’-GG-3’ at T-arm
		5	5’-UU-3’
E	tRNA^Val^-UAC (ANC-loop)	1	Expanded 9-nt loop
		2	Typical 7-nt loop
F	tRNA^Leu^-UAA (ANC-loop)	1	Expanded 9-nt loop
		2	Typical 7-nt loop
G	tRNA^Leu^ -CAA (V-loop)	1	5’-AAAG -3
		2	5’-CAAG-3′
		3	5’-AAAC-3′
		4	5’-AAAU-3’
H	tRNA^Ser^ -GGA (V-loop)	1	5’-UUUU-3’
		2	5’-GUUU-3’

## Data Availability

Not applicable.

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
