# Peer review of "Structural Diversities and Phylogenetic Signals in Plastomes of the Early-Divergent Angiosperms: A Case Study in Saxifragales"

_plants, 2022, doi:10.3390/plants11243544_

Round 1
Reviewer 1 Report
In this study, Han et al. carried out an extensive analysis of plastome data (208 sequences) within the order Saxifragales to analyse structural differences in this organelle, useful for phylogenetic or, in general, evolutionary studies. In general, the manuscript is well written and the data have been analysed thoroughly in structural, comparative and phylogenomic analyses. Some sections should be checked for English and some sentences rephrased for a better comprehension (especially in the abstract). I am providing a list of comments to help in these tasks. After these adjustments, I am confident that the paper is ready for publication.
15 remove “a” before representative; and comprises 15 families
17 you can merge (and shorten) the two sentences as follows: “our previous studies found significant structural diversities among the plastomes of several lineages, suggesting a possible role in elucidating deep phylogenetic relationships”
18 this line should better state your goals and methods. (e.g., in this study, we collected information from 208 plastomes to…). Please also specify what data are new and what are from literature.
22 please use a better synonym for “diversities”
23 please clarify here what is the meaning of the acronym pttRNA (it’s the first time it is used)
34 what do the author mean for “can hardly be ignored”? Plastid data play an important role in plant systematics, please reformulate the sentence
42 plastome data
43 remove “and” before “to date”
74 “is monophyletic” or “was found to be monophyletic”
80 if these considerations are from published studies, it must be specified
92 please simply state “our findings” or “in this study”
112 please remove “and” at the beginning of the sentence
136 please check “that with only”
137 As listed in Table S2; probably determined (unless intended as “specific”) should be removed
139 please explain the meaning of the acronym HPR
148 as outlined in Figure 1b…
151 “rather” can be removed
155 this sentence is for discussion
158 please check the verb use, there are two verbs in the sentence “There were totally 432 indel events have been obtained”; please rephrase.
160 what do the authors mean for microstructural mutations?
161 varieties? Please check
169 probably rather than DNA barcoding at family level, this pattern of variation of indels (if phylogenetically informative) could be useful for assessing phylogenetic relationships at family level. Unless working with processed material, taxa can be easily recognised at family level with morphological keys.
173 remove “be”
184 please remove the comma after pttRNAs and add “and”; replace “noted” with “indicated” or “marked”
198 rather than diversifications is differences, heterogeneity
200 please replace “for” with “to”
206 differences, not variations
225 please specify the meaning of the acronym PCGs
227 phylogenomic
234 groups? Instead of alliances
243 please use “indicate” instead of “note”
249 with the exclusion of
266 “of” instead of “for”
266-268 please reformulate this sentence including the comments I have outlined above; furthermore, “accessing” is not the right verb;
272 accelerate the knowledge?
274 loss events are not rare across their evolutionary history…
275 please check the sentence, there is something missing between “Jansen et al. [14] sampled 77 PCGs and four rRNAs from each of 64 plastomes, 276 representing the most major angiosperms groups” and “this study suggested…”
326 please replace “from” with “by”
334 “involved”, not “organized”
336 act as
398 many, not abundant
442 could you please be more detailed? Do the authors mean that they re-analyzed the plastome data?
448 please provide details of trimming
476 How was the evolutionary model chosen?
478 A Bayesian phylogenetic analysis was carried out…
480 How frequently were the chains sampled?
485 “among them” is not needed
497 of, not for
499 please remove “the” before phylogenetic tree
In Table S1 it is not clear what the values in columns refer to and what is their unit (%? Length?). Please provide this information after the title and in the text.
Table S2 = please remove “the” and “that” in the title; please check the marker names for letters in italics and regular style. Same for Table S4. Accordingly, please change the titles at line 499
In the reference list scientific names or genera epithets are never in italics; please edit them.
In the whole manuscript, please check the marker names for letters in italics and regular style
Reviewer 2 Report
Review of the article by Shiyun Han et al. entitled "Structural diversities and phylogenetic signals in plastomes of the early-divergent angiosperms: a case study in Saxifragales".
This manuscript deals with analysis of plastome data to study phylogenetic relationships within Saxifragales. Considering various structural rearrangements in plastomes, the authors discuss the possibility of their use as phylogenetic markers. The results obtained are valuable and expand our knowledge about structural organization of plastome in Saxifragales. The manuscript is generally well written and worthy of publication. However, the article cannot be accepted in its present form and requires revision before final acceptance.
The most significant drawback of this article is the illustrative material.
Figure 1c shows a taxon sampling used in the analysis. However, it remains unclear how the number of plastomes used correlates with the number of species and genera in each family, and how these numbers correlate with the size of the families as a whole.
In the Figure 2a, it is necessary to clarify in which species the presented features are found. It is also necessary to mark the name of each stem and loop in the secondary structures that are mentioned in the text.
In the Figure 2b, it is not clear which parts of the secondary structure are affected by the observed changes.
Lines 115-126: an illustration is needed to make the text easier to read.
There is no comparison of trees reconstructed from the PCG and CPG matrices. This needs to be included in the text since the authors refer now to one tree (CPG, line 381), then to another tree (PCG, lines 238, 242, 276, 401, 416) during the discussion and presentation of the results.
Line 293: Undoubtedly, the sequence of complete plastid genomes contains much more informative features than individual fragments. However, it is still not correct to put the markers of the nuclear and plastid genomes in one row.
Line 719: Please provide correct citation of Hall et al.
The correctness of the assessment of the states of signs (apomorphic or prelesiomorphic) raises doubts, taking into account the limited number of outgroups.
It is also not discussed how many indels were homoplastic.
Round 2
Reviewer 2 Report
I have reviewed this manuscript previously and I am happy to say that the authors have improved their manuscript. I have only some minor concerns.
Lines 128-141: As in the previous review, I still advise the authors to add an illustration to make the text easier to read. Moreover, structural rearrangements at the boundaries are described too simply; there is no information about the sizes of extensions or contractions. The authors report that such structural changes as expansions or contractions of inverted repeats have taxonomic significance (lines 68-73), but do not discuss their potential in relation to the Saxifragales.
There is no comparison of trees reconstructed from the PCG and CPG matrices. I propose to transfer some information from the authors' response (Response 5) to the text of the article.
